# Impact of social distancing on the spread of common respiratory viruses during the coronavirus disease outbreak

**Min-Chul Kim[1], Oh Joo Kweon[2], Yong Kwan Lim[2], Seong-Ho Choi ᴼ[1]*, Jin-Won Chung[1], Mi-Kyung Lee[2]**

1 Division of Infectious Diseases, Department of Internal Medicine, Chung-Ang University Hospital, Chung-Ang University College of Medicine, Seoul, Republic of Korea, 2 Department of Laboratory Medicine, Chung-Ang University Hospital, Chung-Ang University College of Medicine, Seoul, Republic of Korea

* tobeservant@cau.ac.kr

**Data Availability Statement:** Korean center for disease control and prevention homepage (http://www.kdca.go.kr/board/board.es?mid=

## Abstract

During the coronavirus disease (COVID-19) pandemic, social distancing was effective in controlling disease spread across South Korea. The impact of national social distancing on the spread of common respiratory virus infections has rarely been investigated. We evaluated the weekly proportion of negative respiratory virus polymerase chain reaction (PCR) test results and weekly positive rates of each respiratory virus during the social distancing period (10th–41st weeks of 2020) and the corresponding period in different years, utilizing the national respiratory virus surveillance dataset reported by the Korean Center for Disease Control and Prevention. The proportions of negative respiratory virus PCR test results increased up to 87.8% and 86.1% during level 3 and level 2 of the social distancing period, respectively. The higher the level of social distancing, the higher the proportion of negative respiratory virus PCR test results. During the social distancing period, the mean weekly positive rates for parainfluenza virus, influenza virus, human coronavirus, and human metapneumovirus were significantly lower than those during the same period in 2015–2019 (0.1% vs. 9.3%, $P$ <0.001; 0.1% vs. 7.2%, $P$ <0.001; 0.4% vs. 2.3%, $P$ <0.001; and 0.2% vs. 5.3%, $P$ <0.001, respectively). The mean positive rate for rhinovirus/enterovirus during level 3 social distancing was lower than that in the same period in 2015–2019 (8.5% vs. 19.0%, $P$ <0.001), but the rate during level 1 social distancing was higher than that in the same period in 2015–2019 (38.3% vs. 19.4%, $P$ <0.001). The national application of social distancing reduced the spread of common respiratory virus infections during the COVID-19 pandemic.

## Introduction

Coronavirus disease 2019 (COVID-19) is caused by severe acute respiratory syndrome coronavirus 2 (SARS-CoV-2). It emerged in Wuhan, China, in December 2019 and evolved into the most devastating pandemic since the 1918 Spanish flu, spreading worldwide in just a few months [1]. Since there is no vaccine or effective antiviral therapy against COVID-19, Korea,

a30502000000&bid=0032). Accessed 12 June 2020.

**Funding:** This work was supported by a National Research Foundation of Korea grant funded by the Korean Government (2019R1C1C006417). The funders had no role in study design, data collection, and analysis, decision to publish, or preparation of the manuscript.

**Competing interests:** The authors have declared that no competing interests exist.

one of the countries closest to China, responded by early detection and isolation of infected patients and by the national application of social distancing [2]. These preventive measures have contributed significantly to stabilizing the outbreak in a few months by dramatically reducing the number of newly infected cases [3]. Although this reduction did not represent the end of the outbreak, the number of critically ill patients reduced concomitantly, preventing the collapse of the medical system and the occurrence of a large number of deaths. Of these measures, social distancing is considered to help prevent the spread of various infectious diseases, including influenza and COVID-19, since it fundamentally reduces contact among people [4, 5]. To prepare for future pandemics caused by various pathogens, it is necessary to evaluate the impact of national social distancing not only on the spread of COVID-19, but also on the transmission of various infectious diseases. In particular, the impact on common respiratory virus infections that may be the cause of ongoing pandemic needs to be examined further. Like other countries in the northern hemisphere, the following eight common respiratory viruses circulate in Korea. Influenza virus (IFV), respiratory syncytial virus (RSV), and human coronavirus (COV) are prevalent in the cold season, human metapneumovirus (MPV) and human bocavirus (BOV) are in spring, and parainfluenza virus (PIV) is in summer. Rhinovirus (RV) shows bimodal peaks that increase in spring and fall, but like adenovirus (ADV), it circulates in all seasons [6, 7]. Therefore, we aimed to investigate the impact of social distancing on the spread of these eight common respiratory virus infections by examining the changes in the rates of these infections before and after the implementation of social distancing for the COVID-19 outbreak. We utilized the national respiratory virus surveillance dataset reported by the Korean Center for Disease Control and Prevention (K-CDC) and a large university hospital dataset containing the data of diagnoses of acute respiratory illnesses. Korea is well known for the strict implementation of social distancing and usage of facemasks compared to that in other countries; thus, it was thought that the effect of social distancing could be clearly observed in Korea.

## Methods

### Data collection

The national respiratory virus surveillance dataset made from Korea Influenza and Respiratory Surveillance System (KINRESS) organized by the K-CDC was retrieved from the K-CDC website [8]. The KINRESS is the public source of laboratory-based epidemiological data showing community outbreaks of influenza and other common respiratory viruses in Korea. Thirty-six primary or secondary clinics throughout the country (recently increased to 52) are providing the K-CDC with respiratory specimens of patients who visit the clinics for acute respiratory illness [6]. Data on the weekly proportion of negative respiratory virus polymerase chain reaction (PCR) test results and the weekly positive rates for each of the eight respiratory viruses were collected between the 1st week of 2015 and 42nd week of 2020. Data on the number of respiratory virus PCR tests performed were available only between the 23rd week of 2017 and the 39th week of 2020 in the K-CDC dataset. The national surveillance data were mainly collected from primary and secondary medical institutions; hence, another dataset from a university hospital, an 850-bed tertiary care teaching hospital in Seoul, was also investigated. All patients who underwent a respiratory virus PCR test between the 1st week of 2015 and the 19th week of 2020 were identified, and data regarding respiratory virus PCR tests (dates and results) were collected at the study hospital. The weekly proportion of negative respiratory virus PCR test results and the weekly positive rates for each of the eight respiratory viruses were also investigated in the hospital dataset. The K-CDC dataset only included data on RV. However, RV is not well distinguished from enterovirus (EV) in commercial respiratory virus

Table 1. Level of social distancing in South Korea.

| Field of application | Level 1 | Level 2 | Level 3 |
|---|---|---|---|
| Overall degree of quarantine | • Daily economic activities allowed while adhering to individual quarantine guidelines | • Staying at home<br>• Minimize unnecessary outings and gatherings<br>• Restrict visits to public facilities | • Staying at home<br>• Restrict all activities other than essential economic activities |
| Individual measures | • Use face masks<br>• Hand washing<br>• Maintaining physical distance between individuals | • Use face masks<br>• Hand washing<br>• Maintaining physical distance between individuals | • Use face masks<br>• Hand washing<br>• Maintaining physical distance between individuals |
| Meetings | • Meetings are allowed | • Indoor meetings with ≥50 people and outdoor meetings with ≥100 people are prohibited | • Meetings with ≥10 people are prohibited |
| Sports events | • Limit the number of spectators | • No spectators | • No sports events permitted |
| Schools | • Allow classes | • Reduce the number of participants in classes<br>• Conduct on-line classes | • Close schools |
| Public facilities | • Allow operation | • Suspend operation | • Suspend operation |
| Private business facilities such as pubs, night clubs, and internet cafes | • Allow operation | • Suspend the operation of high-risk private business facilities | • Suspend the operation of intermediate to high-risk private business facilities |
| Public institutions | • Limit attendance to two-thirds of the normal value<br>• Working from home | • Limit attendance to half of the normal value<br>• Working from home | • Only essential personnel are permitted to go to work |

PCR tests, and the university hospital dataset reported RV/EV in a single category. Thus, the category of RV/EV was uniformly utilized in this study. Data on the number of PCR tests for SARS-CoV-2 and the number of positive test results between the 7th week and 39th week of 2020 were collected from the K-CDC website [9].

## Social distancing

National social distancing officially began on March 22, 2020 (from the 13th week of 2020) [10]. However, public concern and response to the COVID-19 outbreak had already begun before March 22, 2020, because hundreds of cases were being reported every day after over 500 cases were identified on February 28. Accordingly, social distancing was suggested by expert groups such as the Korean Medical Association and the Korean Society of Infectious Diseases through the media from early March. Thus, in this study, the social distancing period was defined from the 10th week of 2020. Details of the levels of social distancing are presented in Table 1. Level 3, level 1, and level 2 social distancing were applied during the 10th–19th, 20th–33rd, and 34th–41st weeks of 2020, respectively.

## Data and statistical analyses

We developed graphs of the weekly number of respiratory virus PCR tests, weekly proportion of negative test results, and weekly number/weekly positive rates for each of the eight respiratory viruses. The mean weekly positive rates of each respiratory virus were compared between the social distancing period (from the 10th week to the 41st week of 2020) and the corresponding periods between 2015 and 2019 (from the 10th week to the 41st week in 2015–2019). Continuous scaled data were compared using the Student's $t$-test, Kruskal-Wallis test, or

Mann-Whitney *U* test. All tests of significance were two-tailed, and a *P* value of <0.05 was considered significant.

### Ethics statement

No ethical approval was obtained, as no patient identifiable information was received and the data were analyzed anonymously.

### Results

The results of the following three PCR tests performed on different patient groups were used in this study– 1) respiratory virus PCR test results were reported by the K-CDC for 303 weeks, but data on the number of performed tests were only available for 176 weeks (35,185 tests, an average of 200 tests per week). 2) respiratory virus PCR tests were performed in 13,045 individuals at the large university hospital in 280 weeks (an average of 47 tests per week). 3) PCR tests for SARS-CoV-2 were performed in 2,426,535 individuals between the 7th and 42nd weeks of 2020 in South Korea (an average of 67,404 tests per week), and 25,129 individuals were diagnosed with COVID-19.

As the number of COVID-19 infections increased rapidly from the 9th week in 2020 and social distancing was implemented from the 10th week, the number of PCR tests decreased by less than a half compared to 2015–2019, and the respiratory virus PCR results was almost negative (Fig 1). The same changes are observed in the university hospital data (S1 Fig).

The more stringent the level of social distancing, the higher was the proportion of negative respiratory virus PCR test results as shown in the left graph of Fig 2 (*P* < 0.001 using the Kruskal-Wallis test; the results of post hoc comparisons using Mann-Whitney *U* test with Bonferroni correction are presented in Table 2). However, this dose-response relationship was not observed between the level of social distancing and the number of respiratory virus PCR test as shown in the right graph of Fig 2, although the number was significantly different between the periods with and without social distancing—adjusted *P* value is <0.001 when comparing the "no social distancing group" and the level 1, 2, and 3 social distancing group, respectively (Table 2).

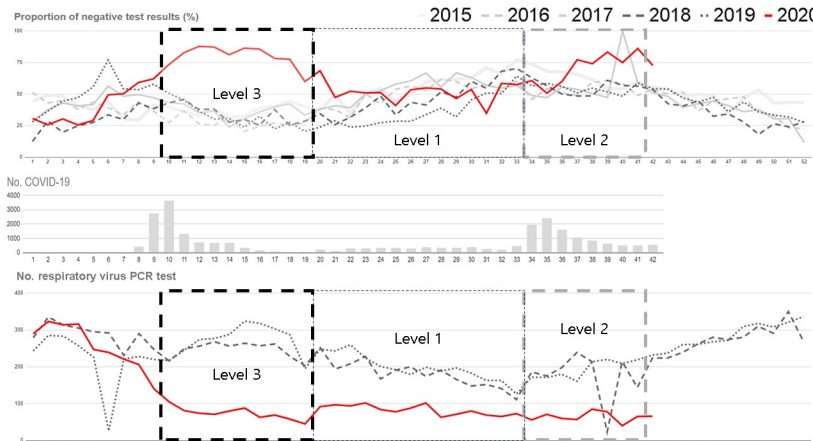

**Fig 1. Weekly proportion of negative respiratory virus PCR test results (upper graph), weekly number of COVID-19 patients (middle graph), and weekly number of respiratory virus PCR tests (lower graph) in South Korea between the 1st week of 2015 and the 42nd week of 2020.** PCR, Polymerase chain reaction; COVID-19, Coronavirus disease 2019.

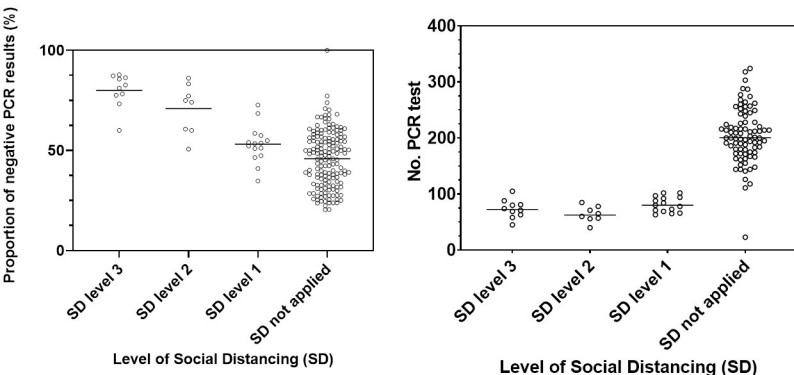

**Fig 2. Mean proportion of negative respiratory virus PCR test results according to the level of social distancing (between the 1st week of 2015 and the 42nd week of 2020; a total of 303 weeks) and the mean number of respiratory virus tests according to the level of social distancing (between the 23rd week of 2017 and the 39th week of 2020; a total of 176 weeks).** The horizontal line represents the mean value, and the hollow circle represents the value for each week. PCR, Polymerase chain reaction.

Each of PIV, IFV, COV, and MPV is rarely observed in the proportion graphs and number graphs in 2020, when social distancing was implemented, unlike 2015–2019 (A-1, A-2, B-1, B-2, C-1, C-2, D-1, and D-2 of Fig 3). In Table 3, the mean positive rates for PIV, IFV, COV, and MPV during the social distancing period were significantly lower than those during the same period in 2015–2019, when social distancing was not implemented (0.1% vs. 9.3%, $P < 0.001$; 0.1% vs. 7.2%, $P < 0.001$; 0.4% vs. 2.3%, $P < 0.001$; and 0.2% vs. 5.3%, $P < 0.001$, respectively). This trend was maintained during each level of social distancing (Table 3). It was difficult to observe the effects of social distancing with respect to RSV positivity in this dataset, because social distancing was implemented outside the general RSV outbreak period (E-1 and E-2 of Fig 3, and Table 3). Although the number of ADV and BOV each appeared to decrease in 2020 compared to 2015–2019 period, there were no differences in positive rates of ADV and BOV between the social distancing period and the corresponding period in 2015–2019 (F-1, F-2, G-1, and G-2 of Fig 3, and Table 3). The proportion of RV/EV decreased during the level 2–3 social distancing period in 2020 compared to 2015–2019, but increased in the level 1 social distancing period (H-1 and H-2 of Fig 3, and Table 3). Graphs of weekly positive rates for the eight respiratory viruses from the university hospital dataset are presented in S2 Fig.

**Table 2. Post hoc comparisons of the proportion of negative respiratory virus PCR test results and the number of respiratory virus PCR tests according to the levels of social distancing (SD) (levels 0, 1, 2, and 3) using the Mann-Whitney $U$ test with Bonferroni correction.**

| Proportion of negative PCR test results | | | No. of PCR tests | | |
|---|---|---|---|---|---|
| Comparison | *P* value | Adjusted *P* value | Comparison | *P* value | Adjusted *P* value |
| SD level 0 vs. 1 | 0.051 | 0.306 | SD level 0 vs. 1 | <0.001 | **<0.001** |
| SD level 0 vs. 2 | <0.001 | **<0.001** | SD level 0 vs. 2 | <0.001 | **<0.001** |
| SD level 0 vs. 3 | <0.001 | **<0.001** | SD level 0 vs. 3 | <0.001 | **<0.001** |
| SD level 1 vs. 2 | 0.004 | **0.024** | SD level 1 vs. 2 | 0.13 | 0.78 |
| SD level 1 vs. 3 | <0.001 | **<0.001** | SD level 1 vs. 3 | 0.255 | 1.53 |
| SD level 2 vs. 3 | 0.083 | 0.498 | SD level 2 vs. 3 | 0.197 | 1.182 |

**NOTE**. SD level 0 presents refers to the years 2015–2019 when SD was not applied.

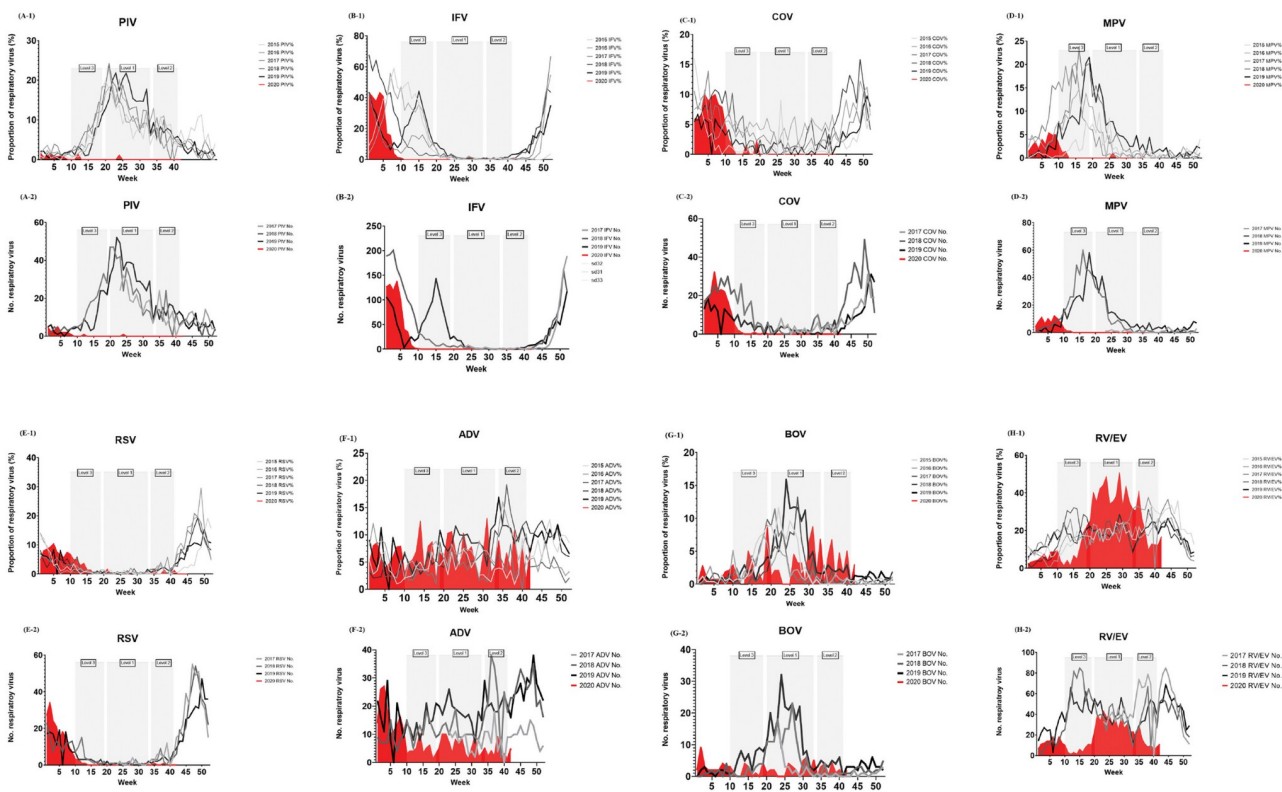

**Fig 3. Weekly number and positive rates of respiratory viruses in South Korea between the 1st week of 2015 and the 42nd week of 2020.** A-1. Proportion of Parainfluenza virus (PIV); A-2. Number of PIV; B-1. Proportion of Influenza virus (IFV); B-2. Number of IFV; C-1. Proportion of Human coronavirus (COV); C-2. Number of COV; D-1. Proportion of Human metapneumovirus (MPV); D-2. Number of MPV; E-1. Proportion of Respiratory syncytial virus (RSV); E-2. Number of RSV; F-1. Proportion of Adenovirus (ADV); F-2. Number of ADV; G-1. Proportion of Human bocavirus (BOV); G-2. Number of BOV; H-1. Proportion of Rhinovirus/enterovirus (RV/EV); H-2. Number of RV/EV. Each of gray squares represents the periods of level 3, level 1, and level 2 social distancing.

## Discussion

The proportion of negative respiratory virus PCR test results increased significantly during the social distancing period in both the K-CDC and hospital datasets. The more stringent the level (levels 3 and 2) of social distancing, the higher was the proportion of negative respiratory virus PCR test results. The weekly positive rates for PIV, IFV, COV, and MPV decreased significantly during the social distancing period. The weekly positive rate of RV/EV also showed a tendency to decrease as the level of social distancing increased, but during level 1 of social distancing, when the intensity of and compliance with social distancing decrease, the rate increased compared to that observed in previous years (when social distancing was not performed).

This study shows that social distancing prevented outbreaks of common respiratory viruses, and this effect was proportional to the level of social distancing. The impact of diverse non-pharmaceutical interventions against COVID-19 has been reported during this pandemic [5]. Additionally, the reduction of the influenza epidemic due to social distancing has already been reported in several studies [11–13]. Common respiratory viruses are transmitted in a manner similar to SARS-CoV-2 and influenza virus. Thus, social distancing is expected to have some effect in terms of suppressing the spread of common respiratory viruses. The results of this study clearly show how extensive social distancing helps prevent the spread of various

**Table 3. Comparison of the mean positive rates for respiratory viruses between the periods with and without social distancing (SD): The Korean Center for Disease Control and Prevention dataset.**

| Respiratory viruses | ADV | PIV | RSV | IFV | COV | RV/EV | BOV | MPV |
|---|---|---|---|---|---|---|---|---|
| Without SD[a] (mean %) | 6.2% | 9.3% | 0.9% | 7.2% | 2.3% | 19.8% | 3.1% | 5.3% |
| With SD[b] (mean %) | 6.2% | 0.1% | 0.5% | 0.1% | 0.4% | 24.5% | 2.6% | 0.2% |
| P value | 0.906 | **<0.001** | **0.046** | **<0.001** | **<0.001** | 0.093 | 0.432 | **<0.001** |
| Without SD3[c] (mean %) | 4.6% | 5.9% | 1.3% | 17.8%[c] | 3.9% | 19.0% | 2.5% | 12.6% |
| With SD3[d] (mean%) | 6.1% | 0.1% | 1.1% | 0.1% | 1.3% | 8.5% | 2.3% | 0.6% |
| P value | 0.077 | **<0.001** | 0.672 | **<0.001** | **0.002** | **<0.001** | 0.822 | **<0.001** |
| Without SD1[e] (mean %) | 6.6% | 14.3% | 0.4% | 1.6% | 1.0% | 19.4% | 4.7% | 4.4% |
| With SD1[f] (mean%) | 6.7% | 0.1% | 0.1% | 0.2% | 0 | 38.3% | 2.8% | 0.1% |
| P value | 0.812 | **<0.001** | 0.069 | **<0.001** | **<0.001** | **<0.001** | 0.104 | **<0.001** |
| Without SD2[g] (mean %) | 9.6% | 6.3% | 1.8% | 0.6% | 2.7% | 22.1% | 0.9% | 0.8% |
| With SD2[h] (mean%) | 5.6% | 0 | 0.3% | 0 | 0 | 20.5% | 2.7% | 0 |
| P value | **0.016** | **<0.001** | **0.004** | **<0.001** | **<0.001** | 0.629 | 0.098 | **<0.001** |

[a]10th–41st weeks in 2015–2019,

[b]10th–41st weeks in 2020,

[c]10th–19th weeks in 2015–2019,

[d]10th–19th weeks in 2020,

[e]20th–33rd weeks in 2015–2019,

[f]20th–33rd weeks in 2020,

[g]34th–41st weeks in 2015–2019,

[h]34th–41st weeks in 2020

ADV, adenovirus; BOV, human bocavirus; COV, human coronavirus; IFV, influenza virus; MPV, human metapneumovirus; PIV, parainfluenza virus; RSV, respiratory syncytial virus; RV/EV, rhinovirus/enterovirus

respiratory viruses, even without the use of antiviral drugs or vaccines. Thus, extensive social distancing may be one of the most effective methods to control a pandemic of similar respiratory viruses. In contrast, the proportion of negative respiratory virus PCR test results may be a surrogate marker to identify whether social distancing is being properly implemented. In May 2020, the media focused on the issue of improper implementation of social distancing (good weather for outdoor activities and various holidays). Therefore, the proportion of negative respiratory virus PCR test results decreased to 60% even before the discontinuation of social distancing (Fig 1). Furthermore, the impact of social distancing on the occurrence of common respiratory virus-associated acute illnesses such as pneumonia, acute exacerbation of chronic obstructive lung disease or asthma, and cardiovascular or cerebral vascular events need to be investigated based on the status of the current pandemic.

The weekly positive rates for PIV, IFV, COV, and MPV significantly reduced during the social distancing period compared with those observed during the corresponding periods in previous years (Table 1). This trend can also be confirmed through the graphs of the university hospital dataset presented in S3, S5, S6 and S9 Figs. However, in the K-CDC dataset, the detection of ADV and BOV was not significantly affected by social distancing as the positive rates for these viruses were not significantly reduced during the social distancing period (Table 3). This may be due to the long-term survival of these respiratory viruses in the surrounding environment, which may result in frequent fomite transmission [14, 15], or due to colonization (rather than infection) of the respiratory tract by these respiratory viruses [16]. In the case of RSV, the period of social distancing overlapped with the RSV outbreak reduction period, and the period after the end of the social distancing period did not represent the usual RSV

outbreak period [C-1 of Fig 3 and S4 Fig]. Thus, the effect of social distancing on the spread of RSV remains unknown.

The proportion of negative respiratory virus PCR test results, which was close to 90% during the social distancing period, decreased to 50% within 3 weeks of the discontinuation of level 3 social distancing in the K-CDC dataset. This shows how fast the transmission of respiratory viruses can be recovered again after reducing the level of social distancing. In the case of RV/EV, a peculiar change was noted. Its proportion rises to almost 50% at the end of level 3 of social distancing and remains around 40% during level 1. It decreases only in the latter part of level 2. It is suggested that transmission of RV/EV occurs easily with reduced compliance of social distancing, and this kind of rapid recovery of transmission can be found only in RV/EV. The reason for the rapid recovery of RV/EV transmission is unclear. One explanation is that RV/EV can be transmitted year-round, unlike other respiratory viruses which show seasonal trends. Additionally, in Korea, where the usage of facial masks is widespread regardless of the level of social distancing, transmission through coughing or sneezing is constantly suppressed, but transmission through contact with unwashed hands may increase as the level of social distancing decreases. RV/EV is well known to be easily transmitted via contact with unwashed hands [17].

As COVID-19 became widespread, the number of respiratory virus PCR tests decreased significantly concomitantly with an increase in the number of PCR tests for SARS-CoV-2. Patients with acute respiratory illness were recommended to visit nearby isolated clinics for COVID-19 screening. Therefore, patients who required medical care for acute respiratory illnesses visited the isolated COVID-19 screening clinics rather than regular clinics. In isolated clinics, PCR tests for common respiratory viruses are rarely performed. The cost of PCR testing both for SARS-CoV-2 and common respiratory viruses is high. Thus, many patients with acute respiratory illnesses underwent only SARS-CoV-2 PCR test. Hospital-affiliated laboratories also experienced difficulties in meeting the demands for further PCR tests in addition to PCR tests for SARS-CoV-2. The situation is thought to be the same in clinics from where the K-CDC surveillance data were collected. The concern is that during a pandemic of a certain respiratory virus, other respiratory viruses may also spread simultaneously. Different respiratory viruses may interact with each other or sometimes cause multiple infections in the same patient. The impact of this aspect on clinical features and outcomes still needs to be investigated [18, 19]. Therefore, government support is needed to ensure that tests for all respiratory viruses are not discontinued during the pandemic, at least in hospitals from where the national respiratory virus surveillance data are collected.

It is possible that the proportion of negative respiratory virus PCR test results increased because respiratory virus PCR tests were performed less frequently. However, this possibility is not valid for the following reasons. First, the proportion of negative respiratory virus PCR test results changed in a stepwise manner with increases in the level of social distancing but the number of respiratory virus PCR tests did not (Fig 2). Thus, it is difficult to consider the change in the number of respiratory virus PCR tests as the main cause of the change in the proportion of negative respiratory virus PCR test results. Second, even if the number of respiratory virus PCR tests decreases, it is difficult to explain why the proportions of positive test results for eight respiratory viruses are different from those in previous years. In particular, PIV, IFV, COV, and MPV were hardly detected during the social distancing period, whereas ADV and BOV continued to be detected; in the case of RV/EV, the rate of detection increases for a while compared to those in previous years. These changes may not be explained by the decrease in the number of respiratory virus PCR tests.

In conclusion, the national application of social distancing against COVID-19 additionally reduced the spread of common respiratory viruses. Social distancing is a primary and effective

strategy for preventing the occurrence of future pandemics caused by novel respiratory viruses.

## Supporting information

**S1 Fig. Weekly proportion of negative respiratory virus PCR test results (upper graph) and weekly number of respiratory virus PCR tests (lower graph) in Chung-Ang University Hospital between the 1st week and the 19th week of 2020.** Middle graph shows the weekly number of COVID-19 patients between the 1st week and 39th week of 2020 in South Korea. PCR, Polymerase chain reaction; COVID-19, Coronavirus disease 2019.
(TIF)

**S2 Fig. Weekly proportion of positive results for adenovirus in Chung-Ang University Hospital between the 1st week of 2015 and the 19th week of 2020.**
(TIF)

**S3 Fig. Weekly proportion of positive results for parainfluenza virus in Chung-Ang University Hospital between the 1st week of 2015 and the 19th week of 2020.**
(TIF)

**S4 Fig. Weekly proportion of positive results for respiratory syncytial virus in Chung-Ang University Hospital between the 1st week of 2015 and the 19th week of 2020.**
(TIF)

**S5 Fig. Weekly proportion of positive results for influenza virus in Chung-Ang University Hospital between the 1st week of 2015 and the 19th week of 2020.**
(TIF)

**S6 Fig. Weekly proportion of positive results for human coronavirus in Chung-Ang University Hospital between the 1st week of 2015 and the 19th week of 2020.**
(TIF)

**S7 Fig. Weekly proportion of positive results for rhinovirus/enterovirus in Chung-Ang University Hospital between the 1st week of 2015 and the 19th week of 2020.**
(TIF)

**S8 Fig. Weekly proportion of positive results for human bocavirus in Chung-Ang University Hospital between the 1st week of 2015 and the 19th week of 2020.**
(TIF)

**S9 Fig. Weekly proportion of positive results for human metapneumovirus in Chung-Ang University Hospital between the 1st week of 2015 and the 19th week of 2020.**
(TIF)

## Author Contributions

**Conceptualization:** Min-Chul Kim, Oh Joo Kweon, Seong-Ho Choi, Jin-Won Chung.

**Data curation:** Min-Chul Kim, Oh Joo Kweon, Yong Kwan Lim, Seong-Ho Choi.

**Formal analysis:** Min-Chul Kim, Oh Joo Kweon, Yong Kwan Lim, Seong-Ho Choi.

**Funding acquisition:** Seong-Ho Choi.

**Investigation:** Min-Chul Kim, Oh Joo Kweon, Seong-Ho Choi.

**Methodology:** Min-Chul Kim, Oh Joo Kweon, Yong Kwan Lim, Seong-Ho Choi.

**Project administration:** Seong-Ho Choi.

**Resources:** Yong Kwan Lim, Seong-Ho Choi.

**Software:** Oh Joo Kweon, Yong Kwan Lim, Seong-Ho Choi.

**Supervision:** Seong-Ho Choi, Jin-Won Chung, Mi-Kyung Lee.

**Validation:** Oh Joo Kweon, Yong Kwan Lim, Seong-Ho Choi, Mi-Kyung Lee.

**Visualization:** Min-Chul Kim, Seong-Ho Choi, Jin-Won Chung, Mi-Kyung Lee.

**Writing – original draft:** Min-Chul Kim, Seong-Ho Choi, Jin-Won Chung.

**Writing – review & editing:** Seong-Ho Choi, Jin-Won Chung, Mi-Kyung Lee.

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
