## [Decision Letter · Decision Letter 0]

16 Feb 2021

PONE-D-21-00571

Impact of social distancing on the spread of common respiratory viruses during the coronavirus disease outbreak

PLOS ONE

Dear Dr. Choi,

Thank you for submitting your manuscript to PLOS ONE. After careful consideration, we feel that it has merit but does not fully meet PLOS ONE’s publication criteria as it currently stands. Therefore, we invite you to submit a revised version of the manuscript that addresses the points raised during the review process.

Thanks for your patience in waiting for the feedback from us, as there is a major difficulty in inviting reviewers. I agree with reviewer 1's comment and surely some more details are needed.

We look forward to receiving your revised manuscript.

Kind regards,

Renee W.Y. Chan, Ph.D.

Academic Editor

PLOS ONE

Journal Requirements:

2.We note that the grant information you provided in the ‘Funding Information’ and ‘Financial Disclosure’ sections do not match.

3.Thank you for stating the following in your Competing Interests section: 

"No. "

4. We note you have included a table to which you do not refer in the text of your manuscript. Please ensure that you refer to Table 3 in your text; if accepted, production will need this reference to link the reader to the Table.

Reviewers' comments:

Reviewer's Responses to Questions

**Comments to the Author**

1. Is the manuscript technically sound, and do the data support the conclusions?

Reviewer #1: Partly

2. Has the statistical analysis been performed appropriately and rigorously? 

Reviewer #1: Yes

3. Have the authors made all data underlying the findings in their manuscript fully available?

Reviewer #1: Yes

4. Is the manuscript presented in an intelligible fashion and written in standard English?

Reviewer #1: Yes

5. Review Comments to the Author

Reviewer #1: In this manuscript, Kim and colleagues determined the effects of social distancing on the transmission of eight common respiratory viruses in Korea. Using the data from Korean Center for Disease Control and Prevention and a local University teaching hospital, the authors found an increased negative PCR rate for common respiratory viruses, which was correlated with the levels of practiced social distancing. The authors also examined PCR positive rates for selected respiratory viruses. The rate for parainfluenza virus, influenza virus, human coronavirus, human metapneumovirus was significantly reduced. No significant changes were seen for adenovirus and bocavirus. The positive rate for human rhinovirus/enterovirus was reduced in level 3 but surprisingly increased during the level 1 social distancing.

Overall, this manuscript provides new information which could help us understand the effects of social distancing on the transmission of respiratory viruses. My concerns and comments are listed as following:

Major comments:

1. It is interesting that no significant changes were seen for adenovirus and bocavirus, and the positive rate for human rhinovirus/enterovirus was reduced in level 3 but surprisingly increased in level 1. However, the positive rate itself is not sufficient to interpret and understand the data. The authors should add the number of positive PCR detection in addition to the positive rate in Figure 2 for each virus.

Besides, which age group has the increased human rhinovirus/enterovirus rate?

2. The figures 2 and 3 are placed in the wrong order and didn’t the match the citations in the main manuscript.

3. Page 15, lines 159-169. The authors wrote: “PCR tests for SARS-CoV-2 were performed in

2,426,535 individuals between the 7th and 42nd weeks of 2020 in South Korea (an average of

67,404 tests per week). Why the number of the weekly average test ranged from 100-300 according to Figure 1. The authors need to explain this.

Also, it is not clear if all listed respiratory viruses were examined in addition to SARS-CoV2 for the same patient.

4. The authors may want to add a short summary about the features of the transmission of common respiratory viruses in the introduction section. For instance, when does rhinovirus/enterovirus season occur in Korea?

Minor comments:

1. Rather than using RV throughout the manuscript, please spell it out and use respiratory viruses instead. It may confuse the reader as RV often refers to rhinovirus.

2. Page 11, lines 93-95. Please consider removing this sentence.

3. Is the study hospital the same as the teaching hospital?

4. Page 9, lines 164-165. This sentence is confusing. Since the reason was given in the discussion section (page 22), the authors may want to simply state that the PCR test number decreased from the beginning of COVID.

6. PLOS authors have the option to publish the peer review history of their article (what does this mean?). If published, this will include your full peer review and any attached files.

Reviewer #1: No

---

## [Author Response · Author response to Decision Letter 0]

22 Mar 2021

RESPONSE TO REVIEWER’S COMMENTS

5. Review Comments to the Author

Reviewer #1: In this manuscript, Kim and colleagues determined the effects of social distancing on the transmission of eight common respiratory viruses in Korea. Using the data from Korean Center for Disease Control and Prevention and a local University teaching hospital, the authors found an increased negative PCR rate for common respiratory viruses, which was correlated with the levels of practiced social distancing. The authors also examined PCR positive rates for selected respiratory viruses. The rate for parainfluenza virus, influenza virus, human coronavirus, human metapneumovirus was significantly reduced. No significant changes were seen for adenovirus and bocavirus. The positive rate for human rhinovirus/enterovirus was reduced in level 3 but surprisingly increased during the level 1 social distancing.

Overall, this manuscript provides new information which could help us understand the effects of social distancing on the transmission of respiratory viruses. My concerns and comments are listed as following:

Major comments:

1. It is interesting that no significant changes were seen for adenovirus and bocavirus, and the positive rate for human rhinovirus/enterovirus was reduced in level 3 but surprisingly increased in level 1. However, the positive rate itself is not sufficient to interpret and understand the data. The authors should add the number of positive PCR detection in addition to the positive rate in Figure 2 for each virus.

Answer) Following your instructions, we made new figures of positive PCR detection for eight respiratory viruses. Also, we added some descriptions to help interpret and understand the revised data. 

New figures) Figure 3 

Added descriptions for the figures) Results

Weekly numbers of each of eight respiratory viruses during the periods of social distancing were generally lower than those without social distancing (Fig 3), because less than half of the respiratory virus PCR tests were conducted during the former periods compared to the latter periods. Only in the case of RV/EV during the period of level 1 social distancing, the weekly numbers of RV/EV were similar to those without social distancing [Fig 3 (F-2)], because of much higher positive rates during the level 1 social distancing period [Fig 3 (F-1)], despite the small number of respiratory virus PCR tests (Fig 1).

Besides, which age group has the increased human rhinovirus/enterovirus rate?

Answer) When K-CDC announced data on respiratory viruses on its web page, it did not disclose detailed data classified by age. Prior to this study, we had requested that age-specific data be shared with us, but we did not get the K-CDC’s cooperation. Unfortunately, as the COVID-19 pandemic came, it was more difficult to obtain such data, and we cannot provide additional data at this time. If the data will be available later, it is considered to be a part that must be analyzed.

2. The figures 2 and 3 are placed in the wrong order and didn’t the match the citations in the main manuscript.

Answer) We corrected the figure order problem you pointed out. Also, a table that was not mentioned in the text in the previous our manuscript was also inserted in the text as table 2.

3. Page 15, lines 159-169. The authors wrote: “PCR tests for SARS-CoV-2 were performed in

2,426,535 individuals between the 7th and 42nd weeks of 2020 in South Korea (an average of

67,404 tests per week). Why the number of the weekly average test ranged from 100-300 according to Figure 1. The authors need to explain this.

Also, it is not clear if all listed respiratory viruses were examined in addition to SARS-CoV2 for the same patient.

Answer) The problem is that SARS-CoV-2 PCR and respiratory virus PCR were performed on different patient groups. Since this part was not clearly presented in the results section, causing misunderstandings. The results section was changed to the following.

Changed texts) results section

The results of the following three PCR tests performed on different patient groups were used in this study – 1) respiratory virus PCR test results were reported by the K-CDC for 303 weeks, but data on the number of performed tests were only available for 176 weeks (35,185 tests, an average of 200 tests per week). 2) respiratory virus PCR tests were performed in 13,045 individuals at the large university hospital in 280 weeks (an average of 47 tests per week). 3) PCR tests for SARS-CoV-2 were performed in 2,426,535 individuals between the 7th and 42nd weeks of 2020 in South Korea (an average of 67,404 tests per week), and 25,129 individuals were diagnosed with COVID-19.

We guess you will know why the average of 100-300 tests per week was recorded.

4. The authors may want to add a short summary about the features of the transmission of common respiratory viruses in the introduction section. For instance, when does rhinovirus/enterovirus season occur in Korea?

Answer) Following your instruction, we changed the manuscript like the followings.

Changed texts) Introduction

Of these measures, social distancing is considered to help prevent the spread of various infectious diseases, including influenza and COVID-19, since it fundamentally reduces contact among people [4, 5]. Although social distancing has been reported in the past and during the SARS epidemic [6], it is probably the first time in this century that social distancing has been widely applied across the country. To prepare for future pandemics caused by various pathogens, it is necessary to evaluate the impact of national social distancing not only on the spread of COVID-19, but also on the transmission of various infectious diseases, including COVID-19. In particular, the impact on common respiratory virus infections that may be the cause of ongoing pandemic needs to be examined further. Like other countries in the northern hemisphere, the following eight common respiratory viruses circulate in Korea. Influenza virus (IFV), respiratory syncytial virus (RSV), and human coronavirus (hCoOV) are prevalent in the cold season, human metapneumovirus (hMPV) and human bocavirus (hBoOV) are in spring, and parainfluenza virus (PIV) is in summer. Rhinovirus (RV) shows bimodal peaks that increase in spring and fall, but like adenovirus (ADV), it circulates in all seasons [6, 7]. Therefore, we aimed to investigate the impact of social distancing on the spread of these eight common respiratory viruses (RV) infections by examining the changes in the rates of these infections before and after the implementation of social distancing for the COVID-19 outbreak.

Minor comments:

1. Rather than using RV throughout the manuscript, please spell it out and use respiratory viruses instead. It may confuse the reader as RV often refers to rhinovirus.

Answer) Following your instructions, RV was spelled out as respiratory virus throughout the manuscript, so as not be confused with rhinovirus (RV)

2. Page 11, lines 93-95. Please consider removing this sentence.

Answer) following your instructions, we removed the sentence.

Changed texts) Introduction

Of these measures, social distancing is considered to help prevent the spread of various infectious diseases, including influenza and COVID-19, since it fundamentally reduces contact among people [4, 5]. Although social distancing has been reported in the past and during the SARS epidemic [6], it is probably the first time in this century that social distancing has been widely applied across the country.

3. Is the study hospital the same as the teaching hospital?

Answer) In order not to be confused in this part, the following changes have been made.

Changed texts) Results

The results of the following three PCR tests performed on different patient groups were used in this study – 1) 

respiratory virus PCR test results were reported by the K-CDC for 303 weeks, but data on the number of performed tests were only available for 176 weeks (35,185 tests, an average of 200 tests per week). 2) respiratory virus PCR tests were performed in 13,045 individuals at the studylarge university hospital in 280 weeks (an average of 47 tests per week). 3) PCR tests for SARS-CoV-2 were performed in 2,426,535 individuals between the 7th and 42nd weeks of 2020 in South Korea (an average of 67,404 tests per week), and 25,129 individuals were diagnosed with COVID-19. 

4. Page 9, lines 164-165. This sentence is confusing. Since the reason was given in the discussion section (page 22), the authors may want to simply state that the PCR test number decreased from the beginning of COVID.

Answer) In order not to be confused in this part, the following changes have been made.

Changed texts) Results

Figure 1 presents the curves of the weekly number of respiratory virus PCR tests and the weekly proportion of negative test results in the K-CDC dataset, with the graph of the weekly number of patients with COVID-19, during the study period. As the number of COVID-19 patients surged from the 9th week of 2020, tThe number of RVrespiratory virus PCR tests rapidly decreased from the 9th week of 2020, and from the 10th week to the 42nd week of 2020, the number of respiratory virus PCR tests continued to be ≤100. The proportion of negative test results reached 87.8% in the 12th week of 2020, exceeded 75% during level 3 of social distancing, and decreased by the 19th week.

Thank you for your thoughtful comments !

---

## [Decision Letter · Decision Letter 1]

13 May 2021

PONE-D-21-00571R1

Impact of social distancing on the spread of common respiratory viruses during the coronavirus disease outbreak

PLOS ONE

Dear Dr. Choi,

Thank you for submitting your manuscript to PLOS ONE. After careful consideration, we feel that it has merit but does not fully meet PLOS ONE’s publication criteria as it currently stands. Therefore, we invite you to submit a revised version of the manuscript that addresses the points raised during the review process.

An additional assessment on your manuscript was carried out on top of the original reviewer. Please address the questions as listed.

We look forward to receiving your revised manuscript.

Kind regards,

Renee W.Y. Chan, Ph.D.

Academic Editor

PLOS ONE

Additional Editor Comments:

The manuscript describes the changes in the detection rate of the common respiratory virus infections by a retrospective study during COVID-19 pandemic in South Korea.

Methods>Data collection:

Line 97: Characteristics of the K-CDC data have been described elsewhere. As this is an important piece of information for the reader to understand the nature of such, a brief description would be essential for easy reference. 

Result:

Line 170: Figure 2, are these data points from 2020 only or all years? How about the resolution of the data, is one dot representing the % of a week?

Table 2. Mann-Whitney U test is for the comparison between two groups, do the authors mean the Kruskal-Wallis test? Please also check the proper post hoc test to be applied.

In Figure 3, the legend is not clear (number verse proportion). There are a series of (A-1) (A-2), (B-1) (B-2) in which not each of them was mentioned in the main text, the author might consider showing figures which provide the key messages to help the reader’s understanding. Please either clarify the differences in the text and so as to demonstrate why it is essential to show both the actual number and the percentage. In addition, please consider showing only one of them to make the message clearer for the reader). Please state clearly what statistical test was performed.

Discussion:

Paragraph 1, please state the nature of the respiratory samples, were those from cases with respiratory symptoms or random surveillance? What are the demographics of these samples? Is there any change in the proportion of age, or nature of the subjects who provided the sample for the test? Is there a change in the respiratory virus surveillance sample collection criteria during the COVID-19 pandemic? This determines the validity of the study.

What is special about the level 1 social distancing and therefore the doubled EV/RV rate?

Overall:

The style of writing is like a laboratory report rather than a manuscript, and at the same time, the legend of the figures did not guide the interpretation of the data clearly.

Reviewers' comments:

Reviewer's Responses to Questions

**Comments to the Author**

1. If the authors have adequately addressed your comments raised in a previous round of review and you feel that this manuscript is now acceptable for publication, you may indicate that here to bypass the “Comments to the Author” section, enter your conflict of interest statement in the “Confidential to Editor” section, and submit your "Accept" recommendation.

Reviewer #1: All comments have been addressed

2. Is the manuscript technically sound, and do the data support the conclusions?

Reviewer #1: Yes

3. Has the statistical analysis been performed appropriately and rigorously? 

Reviewer #1: Yes

4. Have the authors made all data underlying the findings in their manuscript fully available?

Reviewer #1: Yes

5. Is the manuscript presented in an intelligible fashion and written in standard English?

Reviewer #1: Yes

6. Review Comments to the Author

Reviewer #1: I carefully read the revised manuscript. The authors have addressed all my questions and I have no further comments.

7. PLOS authors have the option to publish the peer review history of their article (what does this mean?). If published, this will include your full peer review and any attached files.

Reviewer #1: No

---

## [Author Response · Author response to Decision Letter 1]

15 May 2021

RESPONSE TO REVIEWER’S COMMENTS

Additional Editor Comments:

The manuscript describes the changes in the detection rate of the common respiratory virus infections by a retrospective study during COVID-19 pandemic in South Korea.

Methods>Data collection:

Line 97: Characteristics of the K-CDC data have been described elsewhere. As this is an important piece of information for the reader to understand the nature of such, a brief description would be essential for easy reference. 

Answer) Following your instruction, I revised the manuscript 

From)

The national respiratory virus surveillance dataset provided by the K-CDC during the study period was retrieved from the K-CDC website [8]. The characteristics of the K-CDC data have been described elsewhere [6].

To)

The national respiratory virus surveillance dataset made from Korea Influenza and Respiratory Surveillance System (KINRESS) organized by the K-CDC was retrieved from the K-CDC website [8]. The KINRESS is the public source of laboratory-based epidemiological data showing community outbreaks of influenza and other common respiratory viruses in Korea. Thirty-six primary or secondary clinics throughout the country (recently increased to 52) are providing the K-CDC with respiratory specimens of patients who visit the clinics for acute respiratory illness [6].

Result:

Line 170: Figure 2, are these data points from 2020 only or all years? How about the resolution of the data, is one dot representing the % of a week?

Answer) One hollow circle in the figure represents one week during the study period. How many weeks have been included in the figure is described in detail in the figure legends.

From)

Fig 2. Mean proportion of negative respiratory virus PCR test results and the mean number of respiratory virus tests according to the level of social distancing 

To)

Fig 2. Mean proportion of negative respiratory virus PCR test results according to the level of social distancing (between the 1st week of 2015 and the 42nd week of 2020; a total of 303 weeks) and the mean number of respiratory virus tests according to the level of social distancing (between the 23rd week of 2017 and the 39th week of 2020; a total of 176 weeks). The horizontal line represents the mean value, and the hollow circle represents the value for each week. 

Table 2. Mann-Whitney U test is for the comparison between two groups, do the authors mean the Kruskal-Wallis test? Please also check the proper post hoc test to be applied.

Answer) Table 2 is the result of statistical analysis of the data shown in Fig 2. Since there are 4 groups in Fig 2, the overall trend was confirmed through the Kruskal-Wallis test, and the results are shown not in the Table 2, but in Lines 172–173.

Line 172-173)

The more stringent the level of social distancing, the higher was the proportion of negative respiratory virus PCR test results (P < 0.001 using the Kruskal-Wallis test…

Answer) If the data of Fig 2 with 4 groups are compared with each other by 2 groups, a total of 6 comparisons are made. Since the comparison is between 2 groups, the Mann-Whitney U test was used – you can see the 6 comparisons in Table 2. I revised the manuscript to clarify this point.

From)

Figure 2 shows the mean proportion of negative respiratory virus PCR test results and the mean number of respiratory virus PCR tests according to each level of social distancing. The more stringent the level of social distancing, the higher was the proportion of negative respiratory virus PCR test results (P < 0.001 using the Kruskal-Wallis test; the results of post hoc comparisons are presented in Table 2). The number of respiratory virus PCR tests was significantly different between the periods with and without social distancing (P < 0.001 using the Kruskal-Wallis test); however, there were no significant differences according to the level of social distancing in the number of respiratory virus PCR tests. 

To)

Figure 2 shows the mean proportion of negative respiratory virus PCR test results and the mean number of respiratory virus PCR tests according to each level of social distancing. The more stringent the level of social distancing, the higher was the proportion of negative respiratory virus PCR test results (P < 0.001 using the Kruskal-Wallis test; the results of post hoc comparisons using Mann-Whitney U test with Bonferroni correction are presented in Table 2). The number of respiratory virus PCR tests was significantly different between the periods with and without social distancing – adjusted P value is <0.001 when comparing the “no social distancing group” and the level 1, 2, and 3 social distancing group, respectively (Table 2). However, when the three social distancing groups were compared with each other in the number of respiratory virus PCR tests, there was no significant difference (Table 2).

In Figure 3, the legend is not clear (number verse proportion). There are a series of (A-1) (A-2), (B-1) (B-2) in which not each of them was mentioned in the main text, the author might consider showing figures which provide the key messages to help the reader’s understanding. Please either clarify the differences in the text and so as to demonstrate why it is essential to show both the actual number and the percentage. In addition, please consider showing only one of them to make the message clearer for the reader). Please state clearly what statistical test was performed.

Answer) 

- I agree that each graph has not been fully explained in the main text, although there are many graphs in Fig 3. First, the legends in Fig 3 were modified as follows.

From)

Fig 3. Weekly number and positive rates of respiratory viruses in South Korea between the 1st week of 2015 and the 42nd week of 2020

3A. Adenovirus (ADV); 3B. Parainfluenza virus (PIV); 3C. Respiratory syncytial virus (RSV); 3D. Influenza (IFV); 3E. Human coronavirus (COV); 3F. Rhinovirus/enterovirus (RV/EV); 3G. Human bocavirus (BOV); Figure 3H. Human metapneumovirus (MPV). Each of gray squares represents the periods of level 3, level 1, and level 2 social distancing.

To) 

Fig 3. Weekly number and positive rates of respiratory viruses in South Korea between the 1st week of 2015 and the 42nd week of 2020

A-1. Proportion of Adenovirus (ADV); A-2. Number of ADV; B-1. Proportion of Parainfluenza virus (PIV); B-2. Number of PIV; C-1. Proportion of Respiratory syncytial virus (RSV); C-2. Number of RSV; D-1. Proportion of Influenza (IFV); D-2. Number of IFV; E-1. Proportion of Human coronavirus (COV); E-2. Number of COV; F-1. Proportion of Rhinovirus/enterovirus (RV/EV); F-2. Number of RV/EV; G-1. Proportion of Human bocavirus (BOV); G-2. Number of BOV; H-1. Proportion of Human metapneumovirus (MPV); H-2. Number of MPV. Each of gray squares represents the periods of level 3, level 1, and level 2 social distancing.

Answer) Since Fig 3 and Table 3 are helpful to each other in understating the results, it has been modified to mention both the main contents of Fig 3 and Table 3 for each virus. 

From)

Figure 3 presents the curves of the weekly number/weekly positive rates for each of the eight respiratory viruses. In Table 3, the mean weekly positive rates of respiratory viruses were compared between the social distancing period and the corresponding periods of 2015–2019. Additionally, the same comparisons were conducted according to the three levels of social distancing (Table 3). During the social distancing period, the mean positive rates for PIV, IFV, COV, and MPV were significantly lower than those during the same period in 2015–2019, when social distancing was not implemented (0.1% vs. 9.3%, P <0.001; 0.1% vs. 7.2%, P <0.001; 0.4% vs. 2.3%, P <0.001; and 0.2% vs. 5.3%, P <0.001, respectively). This trend was maintained during each level of social distancing. The mean positive rate for RSV during level 2 social distancing was lower than that during the same period in 2015–2019 (0.3% vs. 1.8%, P = 0.004). However, considering both Table 3 and Figure 3, social distancing was implemented outside the general RSV outbreak period; thus, it was difficult to observe the effects of social distancing with respect to RSV positivity in this dataset. In the case of ADV and BOV, there were no differences in mean positive rates between the social distancing period and the same period in 2015–2019 (6.2% vs. 6.2%, P = 0.906 and 2.6% vs. 3.1%, P = 0.432, respectively). In the case of RV/EV, the mean positive rate during level 3 social distancing was lower than that in the same period in 2015–2019 (8.5% vs. 19.0%, P <0.001); however, during level 1 social distancing, the mean positive rate was higher than that in the same period in 2015–2019 (38.3% vs. 19.4%, P <0.001). During level 2 social distancing, the mean positive rate for RV/EV was not different from that in the same period in 2015–2019, but it was observed that the weekly positive rate for RV/EV decreased in the latter part of this period [Fig 3 (F-1)]. Weekly numbers of each of eight respiratory viruses during the periods of social distancing were generally lower than those without social distancing (Fig 3), because less than half of the respiratory virus PCR tests were conducted during the former periods compared to the latter periods. Only in the case of RV/EV during the period of level 1 social distancing, the weekly numbers of RV/EV were similar to those without social distancing [Fig 3 (F-2)], because of much higher positive rates during the level 1 social distancing period [Fig 3 (F-1)], despite the small number of respiratory virus PCR tests (Fig 1). Graphs of weekly positive rates for the eight respiratory viruses from the university hospital dataset are presented in S2 Fig.

To)

Figure 3 presents the curves of the weekly number/weekly positive rates for each of the eight respiratory viruses. In Table 3, the mean weekly positive rates of respiratory viruses were compared between the social distancing period and the corresponding periods of 2015–2019. Additionally, the same comparisons were conducted according to the three levels of social distancing. Each of PIV, IFV, COV, and MPV is rarely observed in the proportion graphs and number graphs in 2020, when social distancing was implemented, unlike 2015-2019 (B-1, B-2, D-1, D-2, E-1, E-2, H-1, and H-2 of Fig 3). In Table 3, the mean positive rates for PIV, IFV, COV, and MPV during the social distancing period were significantly lower than those during the same period in 2015–2019, when social distancing was not implemented (0.1% vs. 9.3%, P <0.001; 0.1% vs. 7.2%, P <0.001; 0.4% vs. 2.3%, P <0.001; and 0.2% vs. 5.3%, P <0.001, respectively). This trend was maintained during each level of social distancing (Table 3). The mean positive rate for RSV during level 2 social distancing was lower than that during the same period in 2015–2019 (0.3% vs. 1.8%, P = 0.004). However, considering both Table 3 and Figure 3 (C-1 and C-2), social distancing was implemented outside the general RSV outbreak period; thus, it was difficult to observe the effects of social distancing with respect to RSV positivity in this dataset. The number of ADV and BOV each appeared to decrease in 2020 compared to 2015-2019 period in the number graphs (A-2 and G-2 of Fig 3), but there was no significant difference between the 2015-2019 period and 2020 in the proportion graphs (A-1 and G-1 of Fig 3). In Table 3, there were no differences in mean positive rates of ADV and BOV between the social distancing period and the same period in 2015–2019 (6.2% vs. 6.2%, P = 0.906 and 2.6% vs. 3.1%, P = 0.432, respectively). In the proportion graph, RV/EV decreased during the level 2-3 social distancing period in 2020 compared to 2015-2019, but increased in the level 1 social distancing period compared to 2015-2019 (F-1 of Fig 3). In the number graph of RV/EV, the number decreased during the level 2-3 social distancing period, but increased to a level similar to that of 2015-2019 during the level 1 social distancing period (F-2 of Fig 3). The mean positive rate of RV/EV during level 3 social distancing was lower than that in the same period in 2015–2019 (8.5% vs. 19.0%, P <0.001); however, during level 1 social distancing, the mean positive rate was higher than that in the same period in 2015–2019 (38.3% vs. 19.4%, P <0.001) (Table 3). Except for RV/EV of the level 1 social distancing period, weekly numbers of each of seven respiratory viruses during the periods of social distancing were generally lower than those without social distancing (Fig 3), because less than half of the respiratory virus PCR tests were conducted during the former periods compared to the latter periods. Graphs of weekly positive rates for the eight respiratory viruses from the university hospital dataset are presented in S2 Fig.

Answer) The actual number graph and the percentage graph were both presented, and not changed to present only one of them. The reason is that only percentage graphs for each virus were presented at the time of initial submission, but it was revised according to reviewer’s recommendation that the interpretation may be unclear without actual number graph. I think the recommendation is appropriate. 

Discussion:

Paragraph 1, please state the nature of the respiratory samples, were those from cases with respiratory symptoms or random surveillance? What are the demographics of these samples? Is there any change in the proportion of age, or nature of the subjects who provided the sample for the test? Is there a change in the respiratory virus surveillance sample collection criteria during the COVID-19 pandemic? This determines the validity of the study.

Answer) 

As mentioned during the revision of the methods section above, K-CDC’s national surveillance system collected samples of patients who visited primary or secondary clinics in Korea with acute respiratory illness. This system started 20 years ago and reports data every week. So, it is difficult to think that there is a significant change in the characteristics of the sample collection and visiting patients, since the data has been steadily collected from the clinical practice for a long time. 

Detailed information on demographics of participating patients cannot be obtained in the data. I replied to the previous reviewer that it was difficult to obtain such additional data due to realistic circumstances. Please refer to the previous “response to reviewers”.

Lastly, there must have been changes in sample collection since the COVID-19 pandemic began. Please refer to this as I have already described this problem in the 5th and 6th paragraphs in the discussion (reduction of the number of patients and the corresponding decrease in the number of PCR tests).

What is special about the level 1 social distancing and therefore the doubled EV/RV rate?

Answer) As for the social distancing level, you can see the difference in Table 1, which is summarized in the main text as follows.

From)

The proportion of negative respiratory virus PCR test results increased significantly during the social distancing period in both the K-CDC and hospital datasets. The more stringent the level (levels 3 and 2) of social distancing, the higher was the proportion of negative respiratory virus PCR test results. The weekly positive rates for PIV, IFV, COV, and MPV decreased significantly during the social distancing period. The weekly positive rate of RV/EV also showed a tendency to decrease as the level of social distancing increased, but during level 1 of social distancing, it increased compared to the rate observed in previous years (when social distancing was not performed). 

To)

The proportion of negative respiratory virus PCR test results increased significantly during the social distancing period in both the K-CDC and hospital datasets. The more stringent the level (levels 3 and 2) of social distancing, the higher was the proportion of negative respiratory virus PCR test results. The weekly positive rates for PIV, IFV, COV, and MPV decreased significantly during the social distancing period. The weekly positive rate of RV/EV also showed a tendency to decrease as the level of social distancing increased, but during level 1 of social distancing, when the intensity of and compliance with social distancing decrease, the rate increased compared to that observed in previous years (when social distancing was not performed). 

Overall:

The style of writing is like a laboratory report rather than a manuscript, and at the same time, the legend of the figures did not guide the interpretation of the data clearly.

Answer) It seems to be because there are many figures and tables, so I could not effectively organize and explain. Hopefully this modification makes it a little better. 

Thank you for your thoughtful comments!

---

## [Editor Report · Decision Letter 2]

20 May 2021

PONE-D-21-00571R2

Impact of social distancing on the spread of common respiratory viruses during the coronavirus disease outbreak

PLOS ONE

Dear Dr. Choi,

Thank you for submitting your manuscript to PLOS ONE. After careful consideration, we feel that it has merit but does not fully meet PLOS ONE’s publication criteria as it currently stands. Therefore, we invite you to submit a revised version of the manuscript that addresses the points raised during the review process.

We look forward to receiving your revised manuscript.

Kind regards,

Renee W.Y. Chan, Ph.D.

Academic Editor

PLOS ONE

Journal Requirements:

Additional Editor Comments (if provided):

The arrangement of the figure legends (they appeared in the middle of the text), and extra lines e.g. row 173 and 193 make things difficult to understand.

The main text is like a report describing the numbers indicated in the tables and figures, but not the usual format of a writing to elaborate research outcome with clear ideas being supported by tables and figure. Authors might consider to restructure the result part.

Discussion, 2nd para, the authors claimed "This is the first time that the effect of social distancing on the outbreaks of common respiratory viruses has been confirmed using national data from a country where social distancing is well implemented." I wonder if this is the case, as similar study was published in other cities, e.g. The Lancet Public Health VOLUME 5, ISSUE 5, E279-E288, MAY 01, 2020 (reference 11 as the authors indicated); and the authors themselves also indicated a similar study as in reference 5.

---

## [Author Response · Author response to Decision Letter 2]

22 May 2021

RESPONSE TO REVIEWER’S COMMENTS

Additional Editor Comments (if provided):

The arrangement of the figure legends (they appeared in the middle of the text), and extra lines e.g. row 173 and 193 make things difficult to understand.

The main text is like a report describing the numbers indicated in the tables and figures, but not the usual format of a writing to elaborate research outcome with clear ideas being supported by tables and figure. Authors might consider to restructure the result part.

Answer) There are too many figures and tables in the results section, so when I tried to explain the in detail, it became very complicated, and it seems that I failed to show the core contents of this study. To improve this, the contents of the results section have been greatly reduced and only the core sentences have been included. The symbols of respiratory viruses in Fig. 3 were also modified in the order of A, B, and C from the first appearing in the main text.

From) The 2nd paragraph of the results

Figure 1 presents the curves of the weekly number of respiratory virus PCR tests and the weekly proportion of negative test results in the K-CDC dataset, with the graph of the weekly number of patients with COVID-19, during the study period. The number of respiratory virus PCR tests rapidly decreased from the 9th week of 2020, and from the 10th week to the 42nd week of 2020, the number of respiratory virus PCR tests continued to be ≤100. The proportion of negative test results reached 87.8% in the 12th week of 2020, exceeded 75% during level 3 of social distancing, and decreased by the 19th week. The proportion of negative test results was maintained around 50% during level 1 social distancing but gradually increased with the start of level 2 social distancing; it increased to 86.1% in the 41st week (Figure 1). A decrease in the number of respiratory virus PCR tests and an increase in the proportion of negative test results during level 3 social distancing were also observed in the university hospital dataset (S1 Fig). 

Fig 1. Weekly proportion of negative respiratory virus PCR test results (upper graph), weekly number of COVID-19 patients (middle graph), and weekly number of respiratory virus PCR tests (lower graph) in South Korea between the 1st week of 2015 and the 42nd week of 2020.

PCR, Polymerase chain reaction; COVID-19, Coronavirus disease 2019

To) The 2nd paragraph of the results 

As the number of COVID-19 infections increased rapidly from the 9th week in 2020 and social distancing was implemented from the 10th week, the number of PCR tests decreased by less than a half compared to 2015-2019, and the respiratory virus PCR results was almost negative (Figure 1). The same changes are observed in the university hospital data (S1 Fig). 

Fig 1. Weekly proportion of negative respiratory virus PCR test results (upper graph), weekly number of COVID-19 patients (middle graph), and weekly number of respiratory virus PCR tests (lower graph) in South Korea between the 1st week of 2015 and the 42nd week of 2020. PCR, Polymerase chain reaction; COVID-19, Coronavirus disease 2019.

From) The 3rd paragraph of the results 

Figure 2 shows the mean proportion of negative respiratory virus PCR test results and the mean number of respiratory virus PCR tests according to each level of social distancing. The more stringent the level of social distancing, the higher was the proportion of negative respiratory virus PCR test results (P < 0.001 using the Kruskal-Wallis test; the results of post hoc comparisons using Mann-Whitney U test with Bonferroni correction are presented in Table 2). The number of respiratory virus PCR tests was significantly different between the periods with and without social distancing – adjusted P value is <0.001 when comparing the “no social distancing group” and the level 1, 2, and 3 social distancing group, respectively (Table 2). However, when the three social distancing groups were compared with each other in the number of respiratory virus PCR tests, there was no significant difference (Table 2). 

Fig 2. Mean proportion of negative respiratory virus PCR test results according to the level of social distancing (between the 1st week of 2015 and the 42nd week of 2020; a total of 303 weeks) and the mean number of respiratory virus tests according to the level of social distancing (between the 23rd week of 2017 and the 39th week of 2020; a total of 176 weeks). The horizontal line represents the mean value, and the hollow circle represents the value for each week. 

PCR, Polymerase chain reaction

To) The 3rd paragraph of the results 

The more stringent the level of social distancing, the higher was the proportion of negative respiratory virus PCR test results as shown in the left graph of Figure 2 (P < 0.001 using the Kruskal-Wallis test; the results of post hoc comparisons using Mann-Whitney U test with Bonferroni correction are presented in Table 2). However, this dose-response relationship was not observed between the level of social distancing and the number of respiratory virus PCR test as shown in the right graph of Figure 2, although the number was significantly different between the periods with and without social distancing – adjusted P value is <0.001 when comparing the “no social distancing group” and the level 1, 2, and 3 social distancing group, respectively (Table 2). 

Fig 2. Mean proportion of negative respiratory virus PCR test results according to the level of social distancing (between the 1st week of 2015 and the 42nd week of 2020; a total of 303 weeks) and the mean number of respiratory virus tests according to the level of social distancing (between the 23rd week of 2017 and the 39th week of 2020; a total of 176 weeks). The horizontal line represents the mean value, and the hollow circle represents the value for each week. PCR, Polymerase chain reaction

From) Table 2

Table 2. Post hoc comparisons of the proportion of negative respiratory virus PCR test results and the number of respiratory virus PCR tests according to the levels of social distancing (SD) (levels 0, 1, 2, and 3) using the Mann-Whitney U test with Bonferroni correction.

Proportion of negative PCR test results No. of PCR tests

Group Comparison P value Adjusted P value Group Comparison P value Adjusted P value

1 SD level 0 vs. 1 0.051 0.306 1 SD level 0 vs. 1 <0.001 <0.001

2 SD level 0 vs. 2 <0.001 <0.001 2 SD level 0 vs. 2 <0.001 <0.001

3 SD level 0 vs. 3 <0.001 <0.001 3 SD level 0 vs. 3 <0.001 <0.001

4 SD level 1 vs. 2 0.004 0.024 4 SD level 1 vs. 2 0.13 0.78

5 SD level 1 vs. 3 <0.001 <0.001 5 SD level 1 vs. 3 0.255 1.53

6 SD level 2 vs. 3 0.083 0.498 6 SD level 2 vs. 3 0.197 1.182

To) Table 2

Table 2. Post hoc comparisons of the proportion of negative respiratory virus PCR test results and the number of respiratory virus PCR tests according to the levels of social distancing (SD) (levels 0, 1, 2, and 3) using the Mann-Whitney U test with Bonferroni correction.

NOTE. SD level 0 presents refers to the years 2015-2019 when SD was not applied.

Proportion of negative PCR test results No. of PCR tests

Comparison P value Adjusted P value Comparison P value Adjusted P value

SD level 0 vs. 1 0.051 0.306 SD level 0 vs. 1 <0.001 <0.001

SD level 0 vs. 2 <0.001 <0.001 SD level 0 vs. 2 <0.001 <0.001

SD level 0 vs. 3 <0.001 <0.001 SD level 0 vs. 3 <0.001 <0.001

SD level 1 vs. 2 0.004 0.024 SD level 1 vs. 2 0.13 0.78

SD level 1 vs. 3 <0.001 <0.001 SD level 1 vs. 3 0.255 1.53

SD level 2 vs. 3 0.083 0.498 SD level 2 vs. 3 0.197 1.182

From) The 4th paragraph of the results 

Figure 3 presents the curves of the weekly number/weekly positive rates for each of the eight respiratory viruses. In Table 3, the mean weekly positive rates of respiratory viruses were compared between the social distancing period and the corresponding periods of 2015–2019. Additionally, the same comparisons were conducted according to the three levels of social distancing. Each of PIV, IFV, COV, and MPV is rarely observed in the proportion graphs and number graphs in 2020, when social distancing was implemented, unlike 2015-2019 (B-1, B-2, D-1, D-2, E-1, E-2, H-1, and H-2 of Fig 3). In Table 3, the mean positive rates for PIV, IFV, COV, and MPV during the social distancing period were significantly lower than those during the same period in 2015–2019, when social distancing was not implemented (0.1% vs. 9.3%, P <0.001; 0.1% vs. 7.2%, P <0.001; 0.4% vs. 2.3%, P <0.001; and 0.2% vs. 5.3%, P <0.001, respectively). This trend was maintained during each level of social distancing (Table 3). The mean positive rate for RSV during level 2 social distancing was lower than that during the same period in 2015–2019 (0.3% vs. 1.8%, P = 0.004). However, considering both Table 3 and Figure 3 (C-1 and C-2), social distancing was implemented outside the general RSV outbreak period; thus, it was difficult to observe the effects of social distancing with respect to RSV positivity in this dataset. The number of ADV and BOV each appeared to decrease in 2020 compared to 2015-2019 period in the number graphs (A-2 and G-2 of Fig 3), but there was no significant difference between the 2015-2019 period and 2020 in the proportion graphs (A-1 and G-1 of Fig 3). In Table 3, there were no differences in mean positive rates of ADV and BOV between the social distancing period and the same period in 2015–2019 (6.2% vs. 6.2%, P = 0.906 and 2.6% vs. 3.1%, P = 0.432, respectively). In the proportion graph, RV/EV decreased during the level 2-3 social distancing period in 2020 compared to 2015-2019, but increased in the level 1 social distancing period compared to 2015-2019 (F-1 of Fig 3). In the number graph of RV/EV, the number decreased during the level 2-3 social distancing period, but increased to a level similar to that of 2015-2019 during the level 1 social distancing period (F-2 of Fig 3). The mean positive rate of RV/EV during level 3 social distancing was lower than that in the same period in 2015–2019 (8.5% vs. 19.0%, P <0.001); however, during level 1 social distancing, the mean positive rate was higher than that in the same period in 2015–2019 (38.3% vs. 19.4%, P <0.001) (Table 3). Except for RV/EV of the level 1 social distancing period, weekly numbers of each of seven respiratory viruses during the periods of social distancing were generally lower than those without social distancing (Fig 3), because less than half of the respiratory virus PCR tests were conducted during the former periods compared to the latter periods. Graphs of weekly positive rates for the eight respiratory viruses from the university hospital dataset are presented in S2 Fig.

Fig 3. Weekly number and positive rates of respiratory viruses in South Korea between the 1st week of 2015 and the 42nd week of 2020

A-1. Proportion of Adenovirus (ADV); A-2. Number of ADV; B-1. Proportion of Parainfluenza virus (PIV); B-2. Number of PIV; C-1. Proportion of Respiratory syncytial virus (RSV); C-2. Number of RSV; D-1. Proportion of Influenza (IFV); D-2. Number of IFV; E-1. Proportion of Human coronavirus (COV); E-2. Number of COV; F-1. Proportion of Rhinovirus/enterovirus (RV/EV); F-2. Number of RV/EV; G-1. Proportion of Human bocavirus (BOV); G-2. Number of BOV; H-1. Proportion of Human metapneumovirus (MPV); H-2. Number of MPV. Each of gray squares represents the periods of level 3, level 1, and level 2 social distancing.

To) The 4th paragraph of the results 

Each of PIV, IFV, COV, and MPV is rarely observed in the proportion graphs and number graphs in 2020, when social distancing was implemented, unlike 2015-2019 (A-1, A-2, B-1, B-2, C-1, C-2, D-1, and D-2 of Figure 3). In Table 3, the mean positive rates for PIV, IFV, COV, and MPV during the social distancing period were significantly lower than those during the same period in 2015–2019, when social distancing was not implemented (0.1% vs. 9.3%, P <0.001; 0.1% vs. 7.2%, P <0.001; 0.4% vs. 2.3%, P <0.001; and 0.2% vs. 5.3%, P <0.001, respectively). This trend was maintained during each level of social distancing (Table 3). It was difficult to observe the effects of social distancing with respect to RSV positivity in this dataset, because social distancing was implemented outside the general RSV outbreak period (E-1 and E-2 of Fig 3, and Table 3). Although the number of ADV and BOV each appeared to decrease in 2020 compared to 2015-2019 period, there were no differences in positive rates of ADV and BOV between the social distancing period and the corresponding period in 2015–2019 (F-1, F-2, G-1, and G-2 of Fig 3, and Table 3). The proportion of RV/EV decreased during the level 2-3 social distancing period in 2020 compared to 2015-2019, but increased in the level 1 social distancing period (H-1 and H-2 of Fig 3, and Table 3). Graphs of weekly positive rates for the eight respiratory viruses from the university hospital dataset are presented in S2 Fig.

Fig 3. Weekly number and positive rates of respiratory viruses in South Korea between the 1st week of 2015 and the 42nd week of 2020

A-1. Proportion of Parainfluenza virus (PIV); A-2. Number of PIV; B-1. Proportion of Influenza virus (IFV); B-2. Number of IFV; C-1. Proportion of Human coronavirus (COV); C-2. Number of COV; D-1. Proportion of Human metapneumovirus (MPV); D-2. Number of MPV; E-1. Proportion of Respiratory syncytial virus (RSV); E-2. Number of RSV; F-1. Proportion of Adenovirus (ADV); F-2. Number of ADV; G-1. Proportion of Human bocavirus (BOV); G-2. Number of BOV; H-1. Proportion of Rhinovirus/enterovirus (RV/EV); H-2. Number of RV/EV. Each of gray squares represents the periods of level 3, level 1, and level 2 social distancing.

Discussion, 2nd para, the authors claimed "This is the first time that the effect of social distancing on the outbreaks of common respiratory viruses has been confirmed using national data from a country where social distancing is well implemented." I wonder if this is the case, as similar study was published in other cities, e.g. The Lancet Public Health VOLUME 5, ISSUE 5, E279-E288, MAY 01, 2020 (reference 11 as the authors indicated); and the authors themselves also indicated a similar study as in reference 5.

Answer) As you pointed out, similar research data have already been reported, so the sentence has been deleted.

This study shows that social distancing prevented outbreaks of common respiratory viruses, and this effect was proportional to the level of social distancing. This is the first time that the effect of social distancing on the outbreaks of common respiratory viruses has been confirmed using national data from a country where social distancing is well implemented. The impact of diverse non-pharmaceutical interventions against COVID-19 has been reported during this pandemic [5]. Additionally, the reduction of the influenza epidemic due to social distancing has already been reported in several studies [11–13]. Common respiratory viruses are transmitted in a manner similar to SARS-CoV-2 and influenza virus. Thus, social distancing is expected to have some effect in terms of suppressing the spread of common respiratory viruses. The results of this study clearly show how extensive social distancing helps prevent the spread of various respiratory viruses, even without the use of antiviral drugs or vaccines. Thus, extensive social distancing may be one of the most effective methods to control a pandemic of similar respiratory viruses. In contrast, the proportion of negative respiratory virus PCR test results may be a surrogate marker to identify whether social distancing is being properly implemented. In May 2020, the media focused on the issue of improper implementation of social distancing (good weather for outdoor activities and various holidays). Therefore, the proportion of negative respiratory virus PCR test results decreased to 60% even before the discontinuation of social distancing (Fig 1). Furthermore, the impact of social distancing on the occurrence of common respiratory virus-associated acute illnesses such as pneumonia, acute exacerbation of chronic obstructive lung disease or asthma, and cardiovascular or cerebral vascular events need to be investigated based on the status of the current pandemic.

Thank you for your thoughtful comments!

---

## [Editor Report · Decision Letter 3]

26 May 2021

Impact of social distancing on the spread of common respiratory viruses during the coronavirus disease outbreak

PONE-D-21-00571R3

Dear Dr. Choi,

We’re pleased to inform you that your manuscript has been judged scientifically suitable for publication and will be formally accepted for publication once it meets all outstanding technical requirements.

Kind regards,

Renee W.Y. Chan, Ph.D.

Academic Editor

PLOS ONE

Additional Editor Comments (optional):

Figure 3: Please make the grey shadow to the back, currently it covers the curve, as it distorted the color of the curves.
---

## [Editor Report · Acceptance letter]

1 Jun 2021

PONE-D-21-00571R3 

Impact of social distancing on the spread of common respiratory viruses during the coronavirus disease outbreak 

Dear Dr. Choi:

I'm pleased to inform you that your manuscript has been deemed suitable for publication in PLOS ONE. Congratulations! Your manuscript is now with our production department. 

Kind regards, 

on behalf of

Dr. Renee W.Y. Chan 

Academic Editor

PLOS ONE